# Fusion Separation of Vanadium-Titanium Magnetite and Enrichment Test of Ti Element in Slag

**DOI:** 10.3390/ma15196795

**Published:** 2022-09-30

**Authors:** Shuangping Yang, Shouman Liu, Shijie Guo, Tiantian Zhang, Jianghan Li

**Affiliations:** School of Metallurgical Engineering, Xi’an University of Architecture and Technology, Xi’an 710055, China

**Keywords:** vanadium titanium magnetite, titanium enrichment, alkalinity, melting temperature, carbon content

## Abstract

In view of the problem that the enrichment and migration law of the Ti element in the slag of vanadium-titanium magnetite during the melting process is not clear, the phase transformation is not clear and the enrichment effect is not obvious, the single factor experiment and orthogonal experiment are used to optimize the melting conditions of Ti enrichment. Through XRD, SEM and EDS analysis, the effects of melting temperature, alkalinity and carbon content on the Ti phase in the slag are studied, and the occurrence form and migration law of the Ti element in the slag system under different melting conditions are clarified. The results demonstrate that increasing the basicity and melting temperature is beneficial to the enrichment of Ti, but it is too high it will lead to the formation of pyroxene, diopside and magnesia-alumina spinel, affecting the enrichment of Ti. The increase in carbon content can make Ti occur in slag in the form of titanium oxides such as TiO, TiO_2_, Ti_2_O_3_ and Ti_3_O_5_, but excessive carbon content leads to the excessive reduction of Ti compounds to TiCN and TiC. After optimization, under the melting conditions of alkalinity 1.2, the melting temperature 1500 °C and carbon content 15%, the content of Ti in slag can reach 18.84%, and the recovery rate is 93.54%. By detecting the content of Fe and V in molten iron, the recovery rates are 99.86% and 95.64%, respectively.

## 1. Introduction

Vanadium-titanium magnetite is a complex symbiotic iron ore resource composed of iron, vanadium, titanium and other metals, which is one of the most important iron ore resources in China [1,2,3,4]. Titanium, as an indispensable metal raw material for the modern industry and advanced science and technology, is mostly derived from the vanadium-titanium magnetite deposits that have been discovered and exploited. How to efficiently separate or enrich titanium from vanadium-titanium magnetite is the premise for its effective utilization. At present, there are mainly two forms of the separation of raw ore and metallurgical enrichment for the utilization of titanium [5].

In the separation of raw ore, Zhang Yanhua and Han Yuexin et al. used weak magnetics to enrich vanadium-titanium magnetite with strong magnetism, and then selected gravity separation, high gradient magnetic separation, electric separation, flotation or their combined processes to obtain titanium concentrate products according to the differences in the properties of weak magnetic tailings [6,7]. Zeng Xiaobo used the “second-stage grinding-second-stage weak magnetic” separation for the vanadium titanomagnetite pre-selection concentrate, and obtained titanium concentrate with a TiO_2_ grade of 46.23% [8]. Li Zhenqian proposed the process of “coarse-grained magnetic pulley dry waste disposal-weak magnetic separation of iron-strong magnetic separation to enrich titanium-reverse flotation desulfurization-flotation purification of ilmenite”, and finally obtained titanium with a TiO_2_ grade of 48.01% concentrate [9]. In terms of the metallurgical enrichment, titanium enters into the slag system to form titanium slag, which is then separated from ferrovanadium. Ma liming et al. milled quicklime and added vanadium-titanium magnetite concentrate to prepare fluxed green pellets. When the basicity of the pellets was 0.6~1.2, the embedding degree and green ball strength of the pellets in the matrix were improved, and then the titanium enrichment was obtained in the slag [10,11]. Zhang Jun et al. [12,13,14,15,16,17,18] adopted the sodium-reduction method to treat vanadium-titanium magnetite, aiming to reduce the melting temperature, realize the enrichment of vanadium, and achieve the effect of vanadium-titanium separation. However, the sodium-enhanced reduction process is easy to cause equipment corrosion and nodules, and malignant expansion hinders the reduction reaction [19,20,21]. Li Juyan et al. obtained high metallization rate pellets at a high temperature through the direct reduction-grinding-separation process, and further broke them, grinded them and then entered magnetic separation to obtain a high-grade metal iron powder and vanadium-rich titanium material. After the wet leaching of vanadium, a titanium-rich material was further obtained [22,23]. Zhao Longsheng et al. used the rotary kiln pre-reduction-electric furnace method to treat vanadium-titanium magnetite, separated iron, vanadium and titanium efficiently, and enriched titanium into slag efficiently [24,25,26]. It can be observed that, whether it is the separation of raw ore or the use of metallurgical processes, there are generally problems such as long process flow, high processing cost, low utilization rate of Ti resources and unclear migration rules of Ti elements.

In this study, vanadium titanomagnetite is used as a raw material, carbon pellets are used internally, and a direct reduction-smelting technology is applied. The disadvantage is high cost. At the same time, compared with the above-mentioned metallurgical Ti enrichment technology, the utilization rate of titanium resources was improved. The efficient enrichment of Ti element in the slag system is realized. After fully considering the matching between the smelting equipment and smelting parameters and materials, practical applications can be realized. In the future research, it can play a certain guiding role for the melting fraction of vanadium titanomagnetite and the enrichment of Ti.

## 2. Materials and Methods

### 2.1. Materials

In this experiment, vanadium-titanium magnetite concentrate from Wuling, Xinjiang was used as raw material, and its chemical composition is depicted in Table 1. It can be observed from Table 1 that the main valuable elements in the raw iron concentrate are Fe, V and Ti, with the mass fractions of TFe, TiO_2_ and V_2_O_5_ are high and equal to 56.20%, 10.89% and 0.74%, respectively. The content of the harmful elements is low and conducive to a comprehensive recycling. It can be observed from Figure 1 that TFe/FeO = 3.09 (<3.5). It can be judged that the raw material is magnetite, because when TFe/FeO = 2.335 in the iron ore raw material is natural pure magnetite, TFe/FeO = 2.335~3.5 is magnetite, and TFe/FeO = 3.5~7.0 is semi-phantom hematite, such as α-Fe_2_O_3_ and γ-Fe_2_O_3_ (maghemite) [27,28]. Figure 1 presents the particle size composition of the vanadium titanium magnetite. The iron concentrate with a particle size below 0.074 mm accounts for 88%, which meets the requirements of granulation. The XRD patterns of vanadium-titanium magnetite are shown in Figure 2. The valuable minerals are mainly hematite (Fe_3_O_4_), ilmenite (FeTiO_3_) and chromite (FeCr_2_O_4_), in addition to a small amount of magnesia spinel (MgFe_2_O_4_), and the diffraction peaks of hematite and magnetite are the most obvious.

The reducing agent used in the test was coke, with the chemical composition given in Table 2. The fixed carbon content of the coke is as high as 84.97%, and the ash, volatile and total sulfur content was low. Its quality met the test requirements. The additives consisted of SiO_2_ and fluorite of analytical-grade and lime with an activity > 200 and a CaO mass fraction of 85%. Adding a certain amount of fluorite CaF_2_ enabled the melting slag system and improved the fluidity of the liquid phase.

### 2.2. Methods

In the experiment, the vanadium-titanium magnetite concentrate powder, 5% coke and 3% bentonite were mixed evenly by a direct reduction and pellet melting process. A PQ10W disc pelletizer was used to prepare the internal carbon pellets. Metalized pellets were obtained by roasting in the CSL-16-12Y high temperature box furnace. After cooling, the pellets were chemically analyzed, the total iron content and metal paste content measured and the metallization rate of the pellets calculated. The reduction melting test of the metalized pellets was carried out in the NBD-M170Q high temperature box furnace and in a graphite crucible as the reaction vessel. The internal carbon metalized pellets, the external coke powder, lime, silica and fluorite were put into the graphite crucible according to a certain proportion, and the cover closed. The pellets were heated in the high temperature box furnace for reducing and melting them. The processing time was 30 min, and the crucible content were taken out and poured immediately into a graphite mold. After cooling, the mold was removed, the samples taken out and the vanadium-bearing pig iron and titanium slag separated. The titanium slag of the reduction product was crushed and grinded below 0.074 mm. The contents of Ti, Fe and V were determined using chemical methods YD2.7.47—91, YD2.8.12—91 and YD2.8.46—91. The test process is shown in Figure 3.

The prepared green balls were metalized by roasting the treatment to further increase the strength and the basicity, with the carbon content controlled through the external distribution of lime and silica and the coke end. The effects of basicity, melting temperature and carbon content on slag phase were explored by controlling different melting temperature. The optimal melting parameters were determined to increase the enrichment of titanium in slag.

## 3. Results and Discussion

### 3.1. Results Analysis of Metalized Pellets

After the moisture was removed from the green ball, the metalized pellets were obtained by roasting at 1250 °C for 60 min. The metallization rate was calculated using Formula (1), and the metallization rate was equal to 73.56%.
(1)$$\gamma =\frac{MFe}{TFe}\times 100\;\%$$

In the formula, *γ* is the metallization rate of pellets (wt %); *TFe* is the total iron mass fraction of the pellets (wt %); *MFe* is the mass fraction of metallic iron in pellets (wt %). The XRD analysis samples were prepared from the metalized pellets, with the results depicted in Figure 4. One can observe that the main phase of the calcined vanadium-titanium magnetite pellets is magnetite (Fe_3_O_4_), and there are also some vanadium-bearing titanium phases, ilmenite (FeTiO_3_), vanadium magnetite (Fe_2_VO_4_) and ilmenite (Fe_0.5_Mg_0.5_Ti_2_O_5_). Due to the insufficient carbon content, Fe mainly exists in the form of Fe_3_O_4_, and iron oxide has not been reduced to metallic iron [29]. The form in which Ti element appears is the same as that in the raw material made of vanadium-titanium magnetite, that is, in the form of ilmenite (FeTiO_3_). At the same time, the metalized pellets still contain the structure of ilmenite, and some of the easily reducible ilmenite crystals has not changed. In the later smelting reduction process, Ti enrichment and Ti and V separation cannot be obtained in the slag. Therefore, the carbon content will continue to increase in the following experiments, and the pellets will be reduced by melting with coke.

### 3.2. Effect of Alkalinity on Slag Phase

The alkalinity was adjusted when the melting temperature was at 1500 °C and the total carbon set at 15%. The increase in alkalinity from 1.0 to 1.1, 1.2 and then to 1.3, which enabled achieving the contents of Ti, Fe and V in the slag, as shown in Figure 5. It is obvious that the content of the Ti element in slag increased first and then decreased with the increase in alkalinity. As for the content of Fe element, it displayed a decreasing trend. During the process of increasing alkalinity from 1.0 to 1.1, the content of Ti increased from 17.42% to 18.91%, and the content of Fe decreased from 3.56% to 1.33%. The reason is that the higher alkalinity is beneficial to the reduction of iron oxide and vanadium oxide, which promotes the migration of Fe and V from the slag phase to iron phase [30]. As shown in Figure 6a,b, the diffraction peak of FeO in the XRD pattern decreased significantly, contrarily to that of TiO_2_. At the same time, the peak in relationship with the Ti_3_O_5_ appeared, revealing the face that increasing alkalinity was beneficial to the enrichment of Ti. When the basicity increased to 1.2, the contents of Ti and Fe in the slag were 19.53% and 0.53%, respectively. As shown in Figure 6c, the Fe oxide presence was almost undetected in the XRD pattern of the slag when the highest peak value of TiO_2_ was reached. This peak corresponded to a Ti-rich compound. At the same time, the peak of the CaSiTiO_5_ phase appeared. With the gradual increase in basicity, it can be observed that the diffraction peak of CaSiTiO_5_ in the slag phase increased, which was mainly because this phase can reduce, in a certain range, the viscosity of a Ti-rich slag. Consequently, the melting temperature of the system also decreased [31]. When the alkalinity increased to a certain range, the area of the liquid phase increased. When the alkalinity increased from 1.2 to 1.3, the Ti content in the slag decreased from 19.53% to 18.96%, and the Fe content decreased from 0.53% to 0.17%. As shown in Figure 6c,d, some low melting point compounds appeared in the melting process: pyroxene (W_1−p_(X,Y)_1+p_Z_2_O_6_), diopside (Ca_2_Mg_X−2X_Si_1−1X_O_7_) and feldspar (CaMgSi_2_O_6_). One can observe that the intensity and number of diffraction peaks corresponding to low melting point compounds increased significantly when the basicity of the material was 1.3. These low melting points improved the fluidity of the slag. This promoted the separation of slag and iron, a process that further reduced the content of Fe [32,33], but the relative content of Ti element was reduced due to the formation of such compounds in the slag. Furthermore, as can be observed from Figure 5, the content of Fe decreased linearly, with that of Ti displaying a downward trend when the alkalinity reached a 1.3 value.

### 3.3. Effect of Melting Temperature on Slag Phase

When the basicity is 1.2 and the total carbon content was equal to 15%, the Ti and Fe contents in titanium slag, at 1450 °C, 1480 °C, 1500 °C, 1520 °C and 1550 °C, were equal to values given in Figure 7. The XRD analysis of slag system is depicted in Figure 8. It can be observed from Figure 7 that the content of Ti in slag increased first and before decreasing with an increase in melting temperature, whereas the content of Fe continuously decreased. When the temperature is increased from 1450 °C to 1500 °C, the Ti and Fe contents changed from 14.18% and 7.36% to 19.46% and 1.36%, respectively. The XRD analysis of slag phase enables noticing that, as shown in Figure 8a–c, for a melting temperature of 1450 °C, the FeO, FeC and Fe_3_C phases are present in the slag; this is demonstrating that the aforementioned temperature failed to provide the energy needed by the reduction of vanadium titanium magnetite. Besides, one observes the diffraction peaks, revealing the presence of a small amount of Fe_3_Ti_3_O_10_ in the analyzed phase. When the melting temperature was increased at 1480 °C, the diffraction peaks of FeO disappear, whereas those of Fe_3_Ti_3_O_10_ appear. When the temperature was further increased at 1500 °C, FeC and Fe_3_C disappeared in the slag, and the amount of Fe_3_Ti_3_O_10_ and Fe_2_TiO_5_ gradually decreased. This reveals that a higher melting temperature is such that more and more iron oxides and titanium iron oxides were reduced, so that more liquid metal iron appeared in the melting phase, and the metal liquid phase gradually increased [34]. This is beneficial to the continuous separation of slag and iron. With the increase in TiO_2_ and the emergence of Ti_2_O_3_, the enrichment form of the Ti element changed from titanium iron oxide to titanium oxide.

When the melting temperature further was increased to 1520 °C and 1550 °C, as shown in Figure 8d,e, the phase composition became complex, and high melting point compounds, such as CaTiO_3_, MgAl_2_O_4_ and Ca_3_Al_2_O_6_, added themselves. This increased the melting temperature affecting the separation of slag and iron and deteriorating the melting effect, which resulted in the further increase in the Fe content and decrease in the slag’s content of Ti. To obtain the best enrichment of the Ti element and the best separation of slag and iron, 1500 °C was selected as the best melting temperature.

### 3.4. Effect of Carbon Content on Slag Phase

When the basicity was 1.2 and the melting temperature set at 1500 °C, the Ti and Fe contents and XRD patterns of slag given by the melting of vanadium-titanium magnetite with total carbon content of 5%, 10%, 15% and 20%, were equated to those depicted in Figure 9 and Figure 10. It can be observed from Figure 9 that Fe and Ti elements achieved a turning point when the carbon content is 15%. When the carbon content was kept at 5%, the content of Ti and Fe in the slag system was 16.66% and 8.36%, respectively. From Figure 10a, it can be observed that the Ti element mainly was in the form of titanium iron oxide, and there was a weak diffraction peak of TiO_2_, when the Fe element was mainly present as iron oxide. It can be observed that phases made of Fe_2_O_3_, FeO and even Fe_3_O_4_ that were present in the raw ore still exist in Figure 10a. This indicates that the carbon content was insufficient for the vanadium-titanium magnetite complete reduction. When the carbon content was further increased at 10%, the vanadium-titanium magnetite was further reduced so that diffraction peaks of iron oxides disappeared in the slag system. The intensity and number of the diffraction peaks of titanium iron oxides decreased, with the peak and number of diffraction peaks of titanium oxides significantly increased. When the carbon content continued to increase up to 15%, the Ti and Fe contents in the slag system reached 19.09% and 1.36%, respectively. The accompanying phase change was such that the number of diffraction peaks of Ti oxides (especially TiO_2_) in the vanadium-titanium magnetite slag system continued to increase, with the type, intensity and number of diffraction peaks of Ti-Fe oxides decreased. The Fe_2_TiO_5′_s diffraction peak gradually decreased, while that of the Fe_3_Ti_3_O_10_ disappeared in the slag system this time.

When the carbon content increased to 20%, the main mineral phase of the vanadium-titanium magnetite slag composition significantly changed. Firstly, Fe_2_TiO_5_ was decreased as well as the type and number of titanium oxide diffraction peaks. At the same time, the number of TiCN’s diffraction peaks significantly increased, and the diffraction peaks, revealing a small amount of TiC, also appeared. This was because too much coke resulted in an over-reduction of TiO_2_ to TiCN; meanwhile, the melting temperature and viscosity of the slag system increased excessively, increasing the fluidity and increasing the iron content. This was a process that resulted in the incomplete separation of slag and iron. This had a negative impact on the reduction-melting process of vanadium-titanium magnetite that hindered the separation process of slag and iron [35]. The titanium-containing phases in the slag system are: black titanium, TiO_2_, Ti_2_O_3_, a small amount of titanium iron oxide, TiCN, TiC and some magnesium aluminum spinel. It can be noticed that the increase in carbon content changed the slag system composition and influenced to a certain extent the thermodynamic and kinetic process of the reduction-melting.

In summary, under the melting conditions of alkalinity of 1.2, with a melting temperature of 1500 °C and a carbon content of 15%, the slag can be better enriched in titanium.

### 3.5. Phase Analysis of Slag

Five groups of melting experiments were carried out by using the melting parameters and process obtained from the above optimization. The chemical composition of the slag phase obtained by melting was analyzed and the recovery rates of each element calculated using Formula (2). The results are shown in Table 3. It can be observed that the slag phase’s content in Ti was 18.84%, the recovery rates of Fe and V in the molten iron were 99.86% and 95.64%, respectively, and the recovery rate of Ti in the slag was 93.54%. Compared with the alkali extraction process, it can be increased by about 4% [36].
(2)$$\gamma =\left(\frac{m_{0}\;\times \;\beta }{m_{1}\;\times \;\alpha }\right)\;\times \;100\%$$

In the formula: γ-element recovery, %; element content in *α*-vanadium-titanium magnetite, %; the content of elements in *β*-iron or slag, %; *m_1_*—Vanadium-titanium magnetite quality, g; *m_0_*—Mass of molten iron or slag, g.

Formula (3) enabled calculating the vanadium distribution ratio between slag and iron. The higher the grade of vanadium in molten iron, the more conducive is the extraction of vanadium into an iron reaction. This means that the smaller ratio is beneficial to the recovery of vanadium. When relying on calculation conducted under the optimized conditions, the vanadium distribution ratio was about 0.26. This enables effectively recovering vanadium from molten iron.
(3)$$L_{V}=\frac{c_{s}\left(V\right)}{c_{m}\left(V\right)}$$

In the formula: *L_V_*—vanadium element distribution ratio, *c_s_* (V)—vanadium grade in titanium slag, %; *c_m_* (V)—Vanadium grade in hot metal, %.

The slag phase obtained by melting and fractionating vanadium titanomagnetite was powdered to obtain the sample that has been subjected to SEM and EDS analyses (Figure 11). From Figure 11a, it can be observed that the titanium slag was in the form of calcium glass. Taking two points A and B from the EDS analysis (see Figure 11b) and of which elemental compositions are consigned in the Table 4, it can be observed from the energy spectrum diffractogram of point A that the diffraction peak of the Ti element is obviously prominent. There is no prominent diffraction peak of the Fe element, indicating that the reduction effect of iron oxide, magnetite and titanomagnetite in the raw material is good. The rest are diffraction peaks of other elements such as O, V, Al, Mg, etc. It can be observed from the element composition that the Ti element at this point is mainly enriched in the form of titanium oxide. As for the point B, it can be observed that the diffraction peaks of Ti and Ca are prominent. This reveals the fact that the Ti element here is mostly enriched in the slag in the form of CaTiO_3_, and there are also diffraction peaks of V, Al, Mg and Si as well as those of other elements. Therefore, under the melting conditions at a melting temperature of 1500 °C, with a basicity of 1.2 and a carbon content of 15%, the Ti element is well enriched in the slag accompanying the melting of vanadium titanite.

## 4. Conclusions

At the end of this study, focusing on the migration law and occurrence state of the Ti element under different conditions during the direct reduction of vanadium titanomagnetite, the following facts deserve to be highlighted:

1. The increase in alkalinity is beneficial to the formation of CaSiTiO_5′_s phase in slag, thereby reducing the viscosity of the slag system, promoting the separation of slag and iron as well as the enrichment of Ti in slag. However, the increase in alkalinity to 1.3 promotes the formation of low melting point compounds such as pyroxene, diopside and feldspar. For this reason, the optimum alkalinity for Ti enrichment is set at 1.2.

2. With the increase in melting temperature, the Ti element in the raw ore is gradually converted by the reduction from FeTiO_3_ to intermediate products (Fe_3_Ti_3_O_10_ and Fe_2_TiO_5_). With the increase in temperature, titanium oxide TiO_2_ and TiO_3_ appear in the slag phase. At 1500 °C, the number of phases in the slag is the least and the enrichment degree of Ti element is the best. With the further increase in temperature, the slag phase’s transformation becomes complex and results in a high melting point—compounds such as magnesium aluminate spinel appear and this is not conducive to the separation of slag and iron and the enrichment of Ti.

3. With the increase in carbon content, the enrichment form of titanium element is reduced from ferrotitanium oxides (Fe_2_TiO_5_ and Fe_3_Ti_3_O_10_) to various titanium oxides (TiO, TiO_2_, Ti_2_O_3_ and Ti_3_O_5_). When the carbon content is greater than 15%, the titanium oxide is over-reduced and TiCN, TiC as well as some magnesia-alumina spinels begin to appear in the slag phase.

## Figures and Tables

**Figure 1 materials-15-06795-f001:**
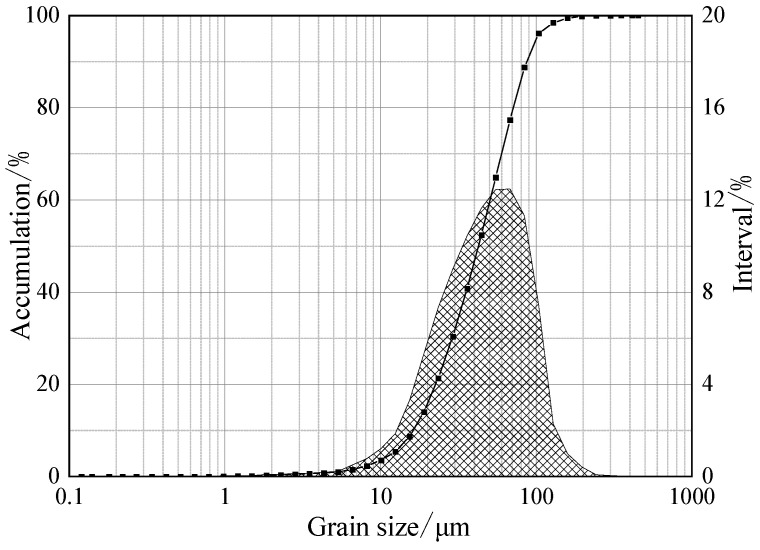
Particle size distribution of vanadium titanium magnetite.

**Figure 2 materials-15-06795-f002:**
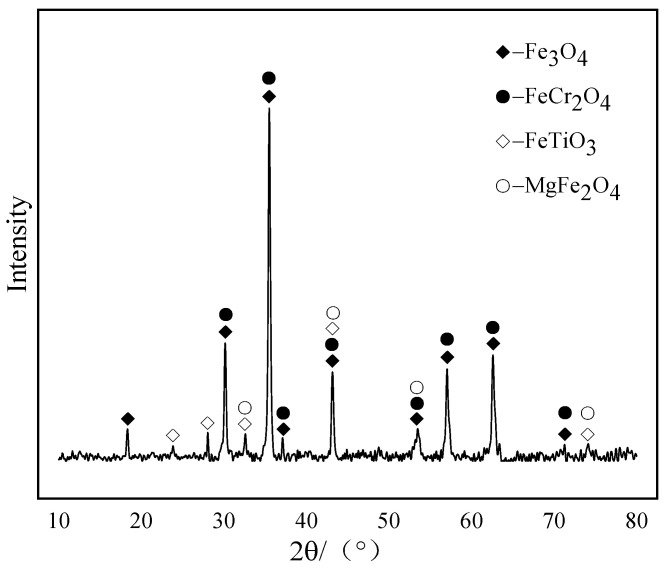
XRD patterns of vanadium titanium magnetite.

**Figure 3 materials-15-06795-f003:**
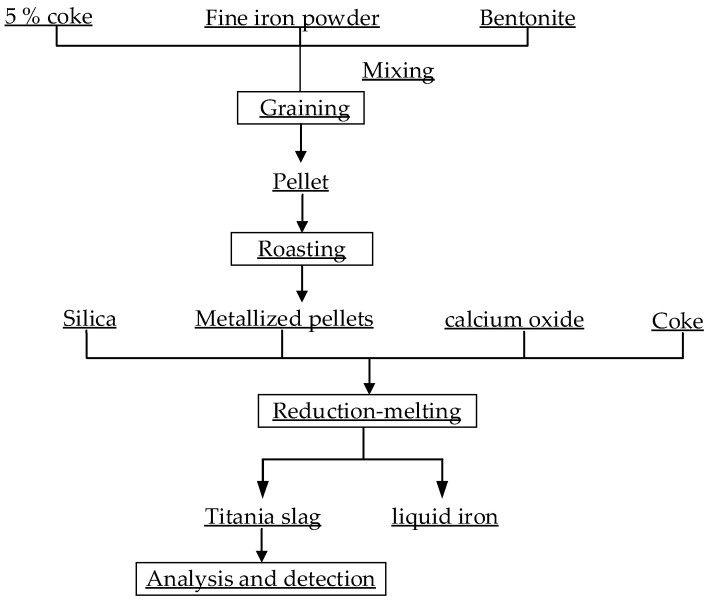
Experimental flow chart.

**Figure 4 materials-15-06795-f004:**
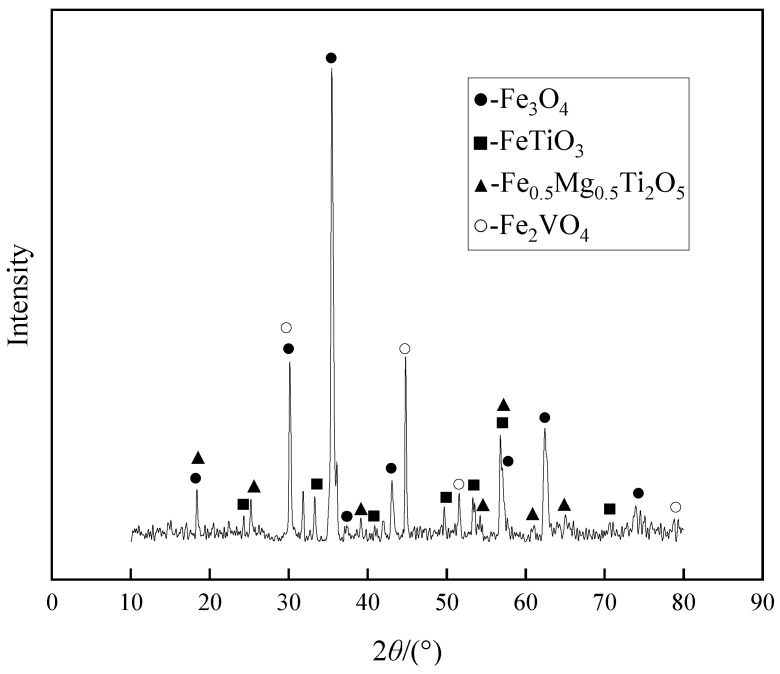
XRD analysis of metalized pellets.

**Figure 5 materials-15-06795-f005:**
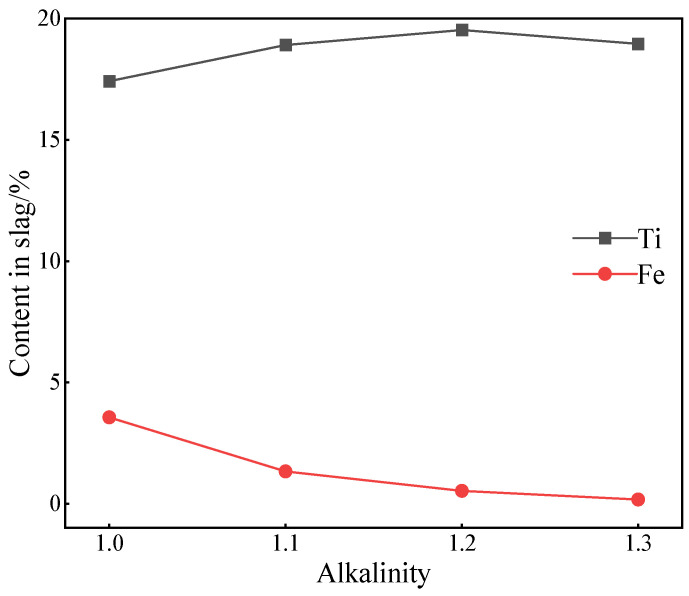
Effect of alkalinity change on Ti/Fe content in slag.

**Figure 6 materials-15-06795-f006:**
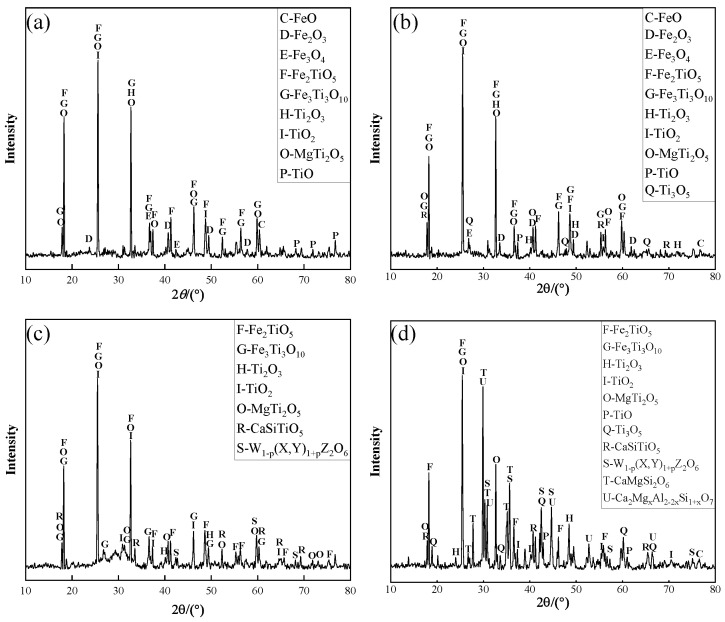
Effect of alkalinity on slag phase XRD. (**a**) Alkalinity = 1.0; (**b**) Alkalinity = 1.1; (**c**) Alkalinity = 1.2; (**d**) Alkalinity = 1.3.

**Figure 7 materials-15-06795-f007:**
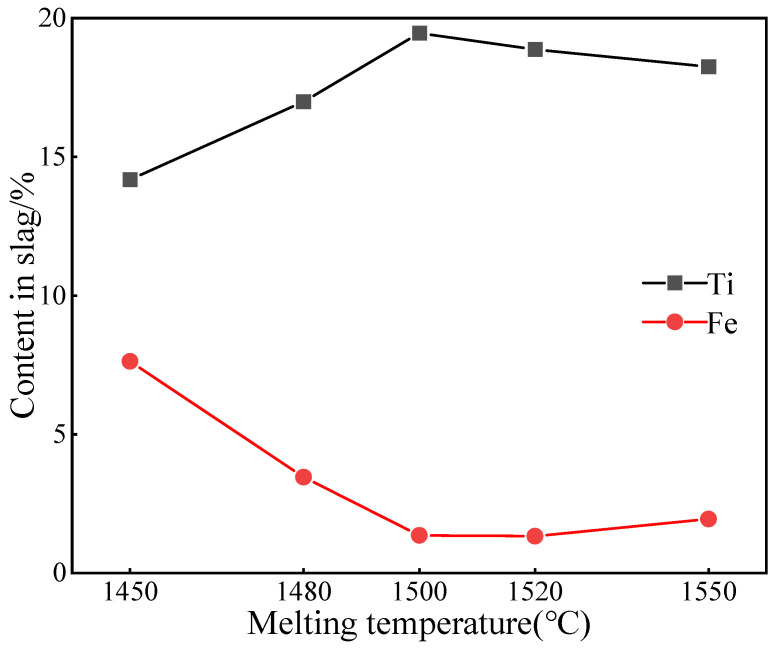
Effect of melting temperature on Ti/Fe content in slag.

**Figure 8 materials-15-06795-f008:**
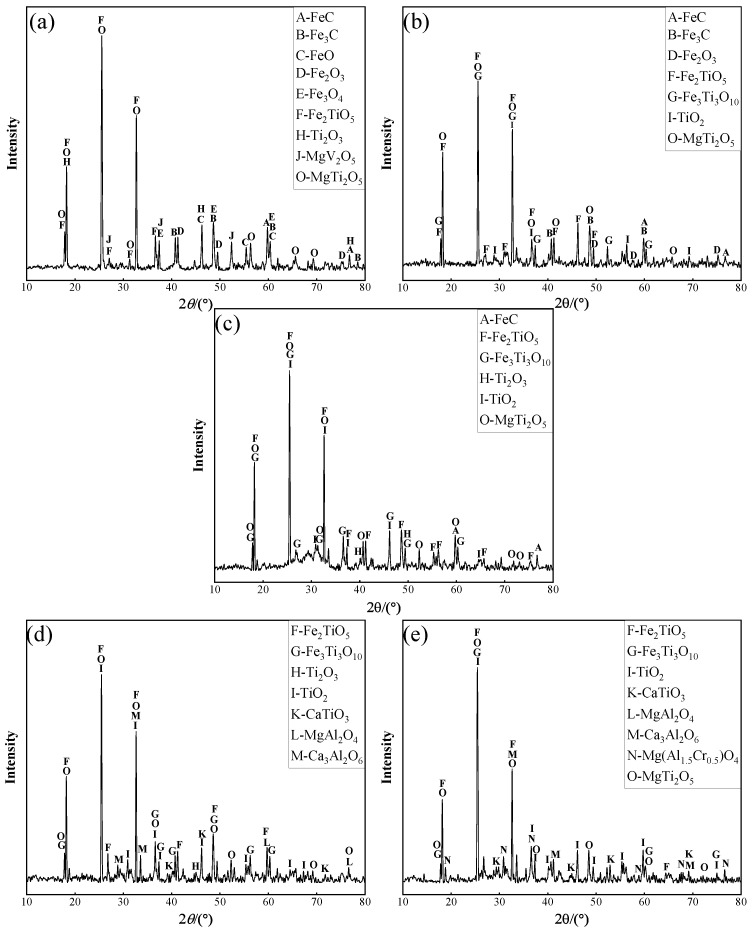
Effect of melting temperature on slag phase XRD. (**a**) 1450 ℃; (**b**) 1480 ℃; (**c**) 1500 ℃; (**d**) 1520 ℃; (**e**) 1550 ℃.

**Figure 9 materials-15-06795-f009:**
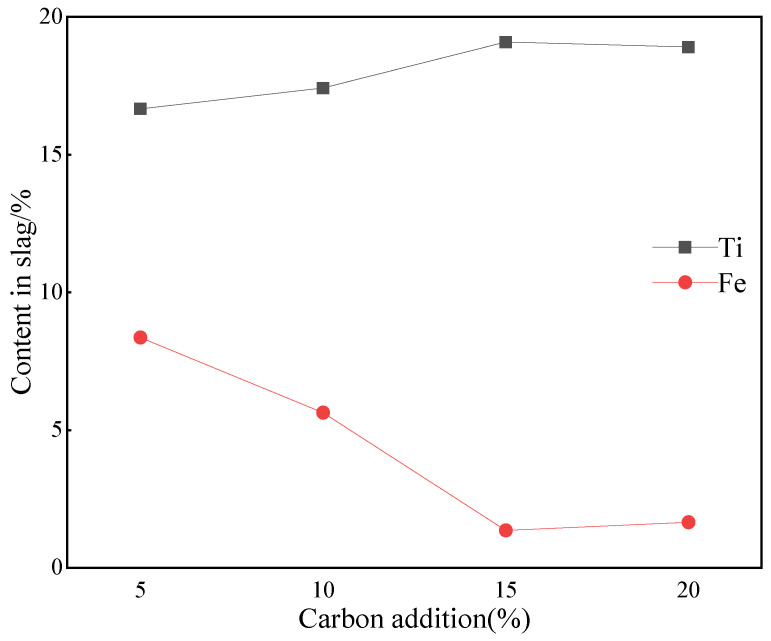
Effect of carbon content on Ti/Fe content in slag.

**Figure 10 materials-15-06795-f010:**
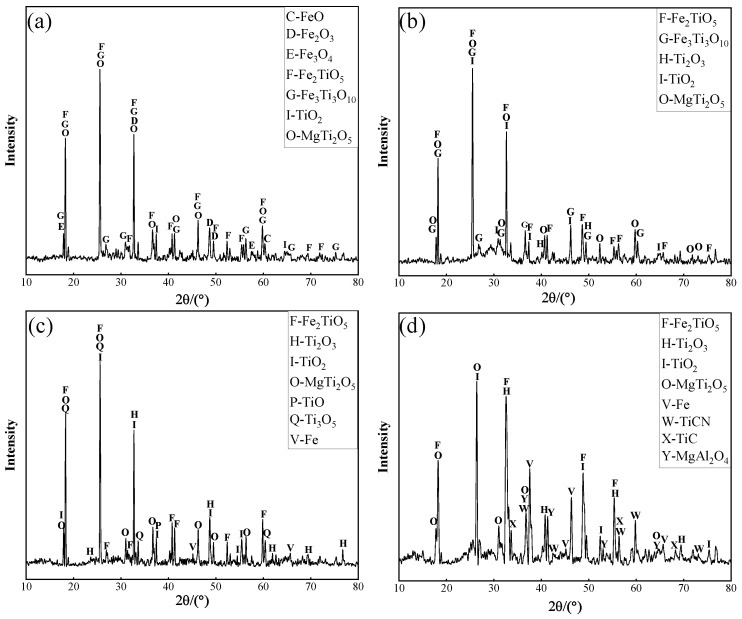
Effect of carbon content on XRD of slag phase. (**a**) Carbon content—5%; (**b**) Carbon content—10%; (**c**) Carbon content—15%; (**d**) Carbon content—20%.

**Figure 11 materials-15-06795-f011:**
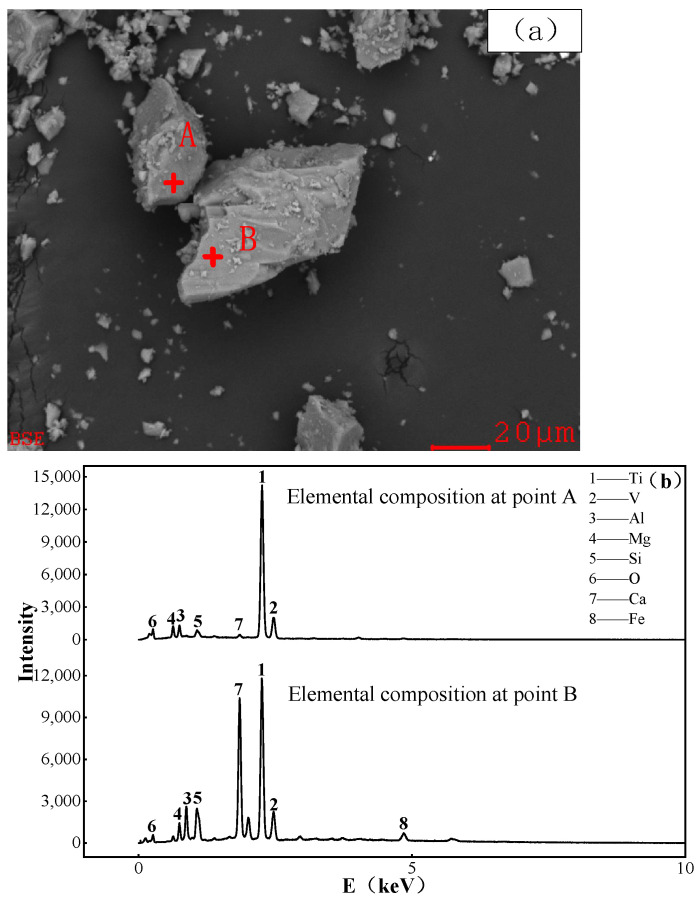
SEM and EDS analysis of slag phase. (**a**) SEM; (**b**) EDS.

**Table 1 materials-15-06795-t001:** Chemical composition of vanadium titanium magnetite concentrate (%).

TFe	FeO	SiO_2_	CaO	MgO	Al_2_O_3_	P	S	TiO_2_	V_2_O_5_	Cr	Burning Loss
56.20	18.19	3.50	0.60	2.30	5.32	0.01	0.03	10.89	0.74	1.08	1.14

**Table 2 materials-15-06795-t002:** Proximate analysis of coke.

Industrial Analysis (%)				Elementary Analysis (%)			Heat (MJ/kg)	Ash Fusion Temperature (℃)			
A_ad_	FC_ad_	M_ad_	V_ad_	S_t,d_	P_d_	H_ad_	Q_net,ar_	DT	FT	HT	ST
13.18	84.97	0.24	1.61	0.32	0.01	0.18	28.03	1120	1210	1170	1150

**Table 3 materials-15-06795-t003:** Chemical composition analysis of slag phase and recovery rate of elements.

Ti	Ca	O	Si	V	Fe	Al	Mg	*γ_Fe_*	*γ_V_*	*γ_Ti_*
18.84	17.82	35.2	17.44	0.13	0.23	7.18	3.16	99.86	95.64	93.54

**Table 4 materials-15-06795-t004:** Composition of point A/B elements.

Element Composition (%)	Ti	V	Al	Mg	Si	O	Ca	Fe
A	54.36	0.51	3.85	4.6932	0.68	34.94	0.70	—
B	41.38	1.17	3.24	1.36	5.60	22.23	21.	3.69

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
