# Peer review of "Fusion Separation of Vanadium-Titanium Magnetite and Enrichment Test of Ti Element in Slag"

_materials, 2022, doi:10.3390/ma15196795_

Round 1

Reviewer 1 Report

Lack of complementary techniques such as Mössbauer spectrometry is observed in the manuscript. Magnetite and maghemite are both similar when studying using XRD then not conclusive data support the presence of magnetite is given by the authors.

Figures (especially diffractograms) are not clear and poor in quality and resolution making difficult to elucidate the envolved phases.

Conclusions must not be enumerated.

Hence, the manuscript has not the enough merits to be published in Materials.

Reviewer 2 Report

·       Figure 3, enlarge the text on the image.

·       Check the bibliography , put it in the format.

·        Show the novelty of the paper compared to the literature, however there is still much work on this topic.

·       Why you choose these materials?

·       In the Introduction section, the last paragraph should contain the scientific contribution and scientific hypotheses of your research. Complete, further elaborate the scientific contribution and scientific hypotheses of your research. Be explicit. In addition to the goal of the research (which was written), the novelty in the context of the scientific contribution should be pointed out. Scientific contributions should be written based on the shortcomings of previous research in the literature. In this way, the authors will better emphasize novelty and scientific soundness.

·       More attention for chemical formulas: example: Fe3C. Correct all the text.

·       Analyze and discuss possibilities of practical application.

·       Why is important the effect of alkalinity on slag phase?

·       Complete the conclusions with the limitations of the proposed methodology. Also write future research.

·       Generally, the quality of the writing could be improved.

Reviewer 3 Report

the article deals with a very interesting topic  and you have garthered very sufficient results of which the discussion enabled achieving a clear understanding of the enrichment issues of titanium and its migration during the alkaline fusion of the raw material. However, After the reviewing process of your article, I have made observations, suggestions and recommendations that deserve to be taken into account for improving the quality of the article. Consequently, I invite you considering my comments attached below

Best regards

Round 2

Reviewer 1 Report

I think the presented manuscript has not been improved significantly to be considered for publication in Materials. Not sufficient methods and techniques to elucidate the involved material's phases are conducted in this work. As previously mentioned Mossbauer technique can be used to correctly elucidate the magnetic iron oxides phases involved and help to understand the whole procedure presented in here, no Rietveld refinement is performed to quantitatively determine the crystalline phases. Only qualitative determination by matching the probable phases was done. I would use diffractogram instead of pattern.

Reviewer 2 Report

Paper was improved.